# Changes in the Diet of an Invasive Predatory Crab, *Chionoecetes opilio*, in the Degrading Benthic Community of an Arctic Fjord

**DOI:** 10.3390/biology13100781

**Published:** 2024-09-29

**Authors:** Alexander D. Kiselev, Anna K. Zalota

**Affiliations:** Shirshov Institute of Oceanology, Russian Academy of Sciences,117997 Moscow, Russia; ad-kiselev@mail.ru

**Keywords:** Decapoda, stomach content analysis, stable isotope analysis, invasive species

## Abstract

**Simple Summary:**

Most of the ecosystems around the world are experiencing high levels of stress. Invasive species are one of the major threats to biodiversity. To successfully prevent and manage this conservation issue, a good understanding of ecological and biological processes involving the native and introduced species is required. This is often challenging due to the interaction of many stressors within an ecosystem, hindering direct changes caused by the invader. Here, we present a unique situation where an invading snow crab is the only disturbing factor in the benthic ecosystem of an arctic fjord. This study looks at how the crabs’ feeding habits change as the invaded ecosystem degrades due to their foraging. Stomach content and stable isotope analyses have revealed that while the prey items have changed, and crabs are forced to feed on less accessible and less nutritional food, their trophic position has not altered. Changes in feeding habits have occurred within the same trophic level. Following the depletion of most nutritional prey items, the number of crabs in the bay has collapsed. We expect a further decrease in crab numbers, a possible partial restoration of the benthic community, and a repeat of the cycle after new settlements of young crabs.

**Abstract:**

The introduction of a new species can lead to substantial changes in a new ecosystem. Local and introduced species’ survival depends on their ability to adapt to the new environment. Studying such adaptations is often hindered by multiple factors affecting the ecosystem. The introduction of a large predatory snow crab, *Chionoecetes opilio*, into the Kara Sea, is a unique invasive species affecting an otherwise undisturbed ecosystem. The crab has caused drastic changes in the macro- and megabenthic taxonomic structure, abundance, and biomass of the most common species in an Arctic fjord, Blagopoluchiya Bay. Stomach content and stable isotope analysis were applied to study crabs’ feeding habits. As the abundance of the most common prey items diminished, the crabs switched to other less accessible food. Prior to substantial changes in benthic communities, the diet of the snow crabs was similar to that of other invaded and native areas, where animal food predominates. However, with the degradation of the ecosystem, detritus contribution has substantially increased. The changes in prey items did not change the crab’s trophic status, and they continued to feed within the same trophic niche. In the depleted benthic communities of Blagopoluchiya Bay, the snow crab is forced to use all available food sources.

## 1. Introduction

The introduction of non-indigenous species into a new environment can lead to drastic changes in the invaded ecosystem [1,2]. Impacts on local communities and their response to the invaders are the subject of numerous studies and are a hot issue in conservation management [3,4,5]. On the other hand, introduced species in a new environment can adapt new life strategies and behaviors to survive and flourish under the new conditions [6]. Often, such plasticity is the key feature of a successful invasive species. After the settlement of a new species, it can reach high numbers and population density, which often leads to the crush of the invaded ecosystem and, consequently, of the invader’s population [7,8,9]. The survival of both local and introduced species is often subject to their ability to adapt to the new environment [10,11,12]. Studying these processes is challenging due to multiple factors that usually affect the ecosystem, such as climate change, anthropogenic pollution, habitat destruction, etc. Therefore, it is a rare opportunity to observe changes in the ecosystem and adaptations of the introduced species unaffected by other factors. Such is the case presented in this study.

The snow crab, *Chionoecetes opilio* (Fabricius, 1788), is an invasive species in the Kara Sea, where it has adversely affected benthic communities [13,14,15] since its introduction. The snow crab’s native range includes the northern Pacific Ocean and the northwestern Atlantic [16,17], where the crab is a well-studied commercial species. It was first found in the Barents Sea in 1996 [18], from where it entered the Kara Sea in 2012 [19]. The crabs have spread throughout the western part of the sea and the bays of the eastern coast of the Novaya Zemlya Archipelago and established a reproducing population [20].

The benthic communities of the Kara Sea have shown undisturbed stability since the first comprehensive studies in the 1920s and 1930s, which were confirmed in 2013 [21,22,23,24,25]. Indeed, due to adverse climate conditions, until recently, this sea experienced low human impact levels, which changed after the sea experienced a slowdown in the formation of sea ice in autumn and its earlier destruction in spring-early summer [26]. Before the snow crab invasion, no large, agile benthic predators existed in the Kara Sea [15,27]. The introduction of a benthic predator into a formerly pristine environment makes this a unique natural experiment, where only the impact of the invader and its adaptation can be observed, not obscured by other stressors [28,29].

Often, marine ecosystems are considered resilient to human-mediated negative impacts [30]. However, this also has limits, especially in less diverse communities [5,31,32]. In comparison to the neighboring Barents Sea, the Kara Sea is characterized by a much lower primary productivity and benthos biomass [27,33,34,35]. In particular, fjords are highly vulnerable to anthropogenic and climate-related impacts due to the lower functional redundancy of macrofauna compared to the open sea [36]. Blagopoluchiya Bay is an Arctic fjord located on the eastern side of the Novaya Zemlya Archipelago in the Kara Sea. It is a long and narrow (15 by 5 km) fjord, with an up to 200 m deep inner basin and a 40 m deep sill separating it from its outer part. The benthic ecosystem of this bay was strongly influenced by the invasion of the snow crabs, resulting in changes in the taxonomic structure, abundance, and biomass of native species [15]. This bay was studied prior to and during the invasion [15,27,37], which makes Blagopoluchiya Bay a unique model fjord to study the invasion of a predator in an otherwise pristine ecosystem.

The negative impacts of predation and food competition are often the most severe [38]. The most drastic changes in the bay’s benthic community and food web structure occurred after 2018, when the crabs reached large enough sizes to feed on the most abundant organisms [15,37]. By 2020–2022, the biomass of previously abundant deposit feeders has crushed, and the most common bivalve suspension feeders diminished, leaving less attractive prey species, such as corals, sea anemones, and sponges [37]. The most dramatic change was probably in the total disappearance of previously dominant brittle stars [15].

After the changes in the benthic ecosystem, the invader’s population collapsed, following the classical invasion curve [7,9]. After the snow crab population reached its maximum impact levels, affecting the surrounding ecosystem, it was forced to adapt to new circumstances. Therefore, it is interesting to see how the crab’s feeding habits caused the changes and how they altered with the degradation of the benthic ecosystem of Blagopoluchiya Bay, leading to its future demise.

The diet of snow crabs is well described in its native range, where it is considered a predatory or omnivorous species, but with the predominance of animal food in its diet [39,40,41,42,43,44]. There are some studies with similar conclusions from its invaded range, including the Kara Sea [45,46,47,48,49,50,51,52,53,54]. Studies from an invasive range suggest a more significant proportion of detritus in its diet than in its natural habitat. Most of the studies are based on the analysis of the contents present in the snow crab’s stomachs.

Stomach content analysis provides insight into what the species consumed directly before being sampled and allows for a quantitative assessment of different prey items. However, the results reflect feeding over a short period, and the quality of prey identification depends on the biology of the studied organisms. Food in the stomachs of decapods is usually highly fragmented due to grinding by the mandibles and the “gastric mill” (calcareous ossicles) in their stomachs [55,56], making it difficult to identify the components. The results may also depend on the characteristics of the prey items: some animals may quickly lose small identifying features due to chewing and digestion, while others may be preserved longer due to hard skeletal elements [57]. The material’s collection speed and fixation also play an important role [58]. Additionally, the results may be distorted by poorly digestible particles in the stomachs, remaining from previous feeding events.

Due to the grinding, the most commonly used quantitate (such as the number of specimens) and weight-based methods to describe species’ diet are hard to apply for decapods. Therefore, to describe the feeding habits of the snow crabs, we use the frequency of occurrence of food components [59,60] and the volume-based method adapted for decapods [60,61] to describe the average (or “virtual”) food lump. Additionally, the overall frequency of dominance, or Tarverdieva’s coefficient (TC) [60], and individual frequency of dominance, showing the frequency of occurrence of stomachs where one of the components occupies more than 60% of the stomach volume, are used. The Froerman’s coefficient (Fr) [60,62] shows the average number of victims in the stomach. These coefficients allow us to ascertain the feeding strategy of decapods according to Burukovsky’s classification [60].

Stable isotope analysis gives more integral information about the feeding preference of a species in time [63,64]. It is a widely used tool to assess marine species’ trophic position and source of organic material [64,65,66]. The carbon isotope ratio changes little across trophic levels and determines the source of organic material (the source of primary production) [67,68]. The nitrogen isotope ratio is typically enriched by 3–4‰ between the prey and the consumer, which allows us to infer their trophic position [68,69,70]. The term “isotopic niche” [71] is broadly used in stable isotope analyses, which describes an area with isotopic δ values as coordinates characterizing the resources and habitat use of the species. These isotopic niches approximate species’ trophic niches [72,73]. Previously, this analysis was used to study the overlap of the trophic niches of the non-indigenous red king crabs [74,75] and snow crabs [54] with other decapods and predatory benthic invertebrates in the Barents Sea.

In this study, we used stomach content and stable isotope analyses to infer changes in the feeding habits of the invasive snow crabs, *Chionoecetes opilio*, in the degrading benthic ecosystem of an Arctic fjord, Blagopoluchiya Bay, under the influence of this invader.

## 2. Materials and Methods

### 2.1. Sample Collection

The snow crabs, *Chionoecetes opilio*, were collected for stable isotope and stomach content analyses at different times of the day in September–October 2018, 2020, and 2022, and only for stomach analysis in 2023 during the R/V cruises of the Shirshov Institute of Oceanology within the framework of a long-term study “Ecosystems of the Siberian Arctic Seas”. The samples were collected at two monitoring stations in Blagopoluchiya Bay. The Inner Basin station is located in the deepest part of the bay (approximately 170 m), and the Sill station is located near the bay’s exit at a depth of 63–72 m (Figure 1). The crabs were collected using a Sigsbee trawl with a frame size of 150 × 35 cm (with an inner mesh of 0.5 cm). Trawl samples were washed through 5 mm and 1 mm mesh sieves. Crabs for stomach analysis were fixed with 10% neutralized formaldehyde as soon as possible after the collection (no more than one hour). All collected crabs were measured (carapace width, CW, accuracy 0.1 mm), and their sex was visually determined.

### 2.2. Stomach Contents Analysis

Stomachs were removed from crabs and stored in 70% alcohol. After dissecting the stomach, its level of fullness was visually assessed. Empty stomachs collected in 2022 and 2023 were preserved and analyzed, while those from 2018 were noted but discarded without being analyzed. The stomach contents were described and analyzed according to the method described by Burukovsky [60]. The contents of each stomach were placed on a Petri dish and covered with a small amount of water. All stomach content objects (SCOs) were analyzed using a binocular stereomicroscope Mikromed MC2 Zoom 1CR. All stomach food items (SFIs; items which may be used as food by the crabs) were identified to the class or order, and if possible, to a lower taxonomic level, using regional taxonomic identification keys [76]. The stomach content objects were measured and described in detail (color, density, structure, etc.), and model examples were photographed.

Frequency of occurrence (FO) was calculated for each stomach content object as the proportion of stomachs where this object was found. An average food lump was calculated as the average volume proportion of each SCO in the generalized food lump to assess the role of SCOs in the snow crab’s diet [60]. For that, the volume fraction of SCOs was visually estimated in full stomachs (70–100% full) (with an accuracy of 10%), and the average volume fraction of each SCO in all full stomachs was calculated. Using this indicator only for full stomachs allows us to avoid errors associated with different digestion rates of SCOs.

In addition, a few more parameters were calculated for the stomach food items (SFIs). The overall dominance frequency, or Tarverdieva’s coefficient (TC), is the frequency of stomachs where one of the food items accounts for more than 60% of the volume (i.e., dominates by volume), calculated taking into account only full stomachs. The individual food item dominance frequency is the frequency of stomachs where this item occupies 60% or more of the volume. Froerman’s coefficient (Fr) is the average number of food items in the stomach, calculated as the sum of all prey item frequencies divided by 100.

### 2.3. Stable Isotope Analysis

The snow crab’s claw muscles were used for stable isotope analysis. The samples were dried in plastic tubes at 70 °C for five days on board the research vessel. The dried material was then crushed using a mortar and pestle and wrapped in tin foil containing 200–500 µg of dried material. Stable isotope analysis was performed in a Thermo Delta V Plus isotopic mass spectrometer and Thermo Flash 1112 Elemental Analyser at the Centre for Collective Use at the Severtsov Institute of Ecology and Evolution, RAS, Moscow, Russia.

Stable isotope composition is reported as ‰ units. Atmospheric N_2_ and Casein pegged to Vienna PeeDee Belemnite (VPDB) were used as standards for nitrogen and carbon, respectively. The analytical error (error of standards) in determining the isotope composition (SD in the laboratory standard analysis of protein B2155, *n* = 4–8) did not exceed 0.2‰ for both δ^13^C and δ^15^N. The minimum, maximum, and mean values of δ^13^C and δ^15^N ± standard deviation (SD) are reported to represent the results.

### 2.4. Data Analysis

The isotopic niche parameters were calculated for crabs of different sexes collected at different stations in different years with a sample size of at least five specimens using the Stable Isotope Bayesian Ellipses in R (SIBER) package [73] using RStudio 2024.04.01 [77,78]. The area of standard ellipses corrected for small sample size (SEA_C_, ‰^2^), which incorporated 40% of the data, was used to compare individual groupings of crabs (by sex, station, and year). Their posterior estimates (SEA_B_) are reported to demonstrate possible variations in SEAc. To maximize possible similarity in feeding habits, the overlap of SEA, including 95% of the data, was used to discuss the isotopic niche proximity (overlap) of crabs. The overlap was reported as the proportion of non-overlapping area (the total overlap area divided by the sum of the areas of two ellipses minus the total overlap area in ‰^2^).

The Kruskal–Wallis test was used to check for differences in the sizes (carapace width, CW), δ^13^C, and δ^15^N between crabs of different sexes, sampling stations, and years. This was followed by the Dunn multiple comparison test with *p* values adjustment using the Holm method. A two-proportion z-test was used to compare frequencies of occurrence, volume proportions of the average food lump (AFL), and individual dominance frequency of stomach content objects in *Chionoecetes opilio* crabs collected from the Sill station in Blagopoluchiya Bay in 2020 and 2023 to the crabs collected in 2018.

## 3. Results

In the four years of study, 321 crabs were sampled (Table 1). All crabs had a hard carapace; therefore, no recently molted specimens were present. Females prevailed in all years, where the female-to-male ratio was 1.5 in 2018 and 2020, and it increased to 3 by 2022 and 2023. Ovigerous females were absent in 2018; some were sampled in 2020 and 2022, but by 2023, most of the sampled females were ovigerous (89 out of 91). As there is one prevailing age group of snow crabs in Blagopoluchiya Bay, the carapace width (CW) mode reflects the most numerous size group sampled in the bay in the respective year. The size of crabs increased from 2018 to 2023 from the mode CW of 27–31 mm to 38–44 mm, respectively. The sizes of crabs were statistically different when different sexes, years, and stations (depth of habitat) were compared (chi-squared = 180.83, df = 10, *p* < 0.01, Table 1). The sizes of crabs from 2020 were not tested due to the small sample size. Generally, crabs from the deep Inner Basin station were larger than those from the Sill station. Most often, males were larger than females. The most significant difference in sizes was between 2018 and 2022 at the Inner Basin station, where there were many crabs with CW less than 40 mm in 2018 and hardly any after 2020. Some specimens in 2018, 2020, and 2023 were used for stomach content and stable isotope analyses.

### 3.1. Stomach Contents

Two hundred ninety-six crabs were dissected for the stomach content analysis (Table 2), from which 260 stomachs were analyzed. Eighty-seven stomachs were full (70–100% full), and the remaining 36 unanalyzed stomachs were empty. However, some empty stomachs were analyzed and revealed that they, too, contained a small portion of items. Further, only the results of analyzed stomachs are reported. The complete list of stomach contents and fullness is reported in the Appendix A. There was a small sample size of analyzed crabs in 2022. Results from these crabs were only used for the overall stomach content objects (SCOs) list (Table 3).

The stomach content objects (SCOs) included mineral particles, detritus, algae and animal fragments, and microplastics (Figure 2, Table 3). Fat droplets were also found in some stomachs. Mineral particles included sand, small stones, and silt. We identified detritus in stomachs as a shapeless, loose organic slurry (Figure 2F). Larger, more defined fragments were attributed to either animal or algae objects. The algae were often represented by numerous small fragments that could not be identified. Some algae fragments were identified as brown algae *Chordaria* sp., *Desmarestia* sp., *Ectocarpus* sp., *Dictyosiphon* sp., and red algae *Ptilota* sp. Diatoms were rare.

Animals were the most diverse group of SCOs. Polychaetes were usually represented by a few fragments and setae (Figure 2B). Some were identified as *Spiochaetopterus* sp. or the *Nephtyidae* family. A few intact bivalve shells were identified as *Nuculana* sp., *Ennucula* sp., *Yoldiella* sp., *Portlandia* sp., and *Ciliacardium* sp. Some fragments were identified only to the Nuculida and Nuculanida family levels, while some small shell fragments were not identified. A cap-shaped shell of a gastropod was found only in one stomach. Crustacean SCO included mysids, shrimps (including *Eualus* sp. and *Pandalus* sp.), and individual fragments of chitin (Figure 2C), as well as fragments of *Chionoecetes opilio* carapace (Figure 2E). Brittle stars (spines and other skeletal elements) were found in the stomachs of crabs in 2018 and 2020 (Figure 2A). Vertebrae or statoliths of fish, nematodes, and bryozoans were less often found in the stomachs. Foraminifera, some identified as *Haplophragmoides* sp., *Dentalina* sp., *Lenticulina* sp., *Elphidium* sp., *Triloculina* sp., *Cornuspira* sp., and *Cibicides* sp. were among the SCO in crabs from the Sill station. No foraminifera were found in crabs from the Inner Basin station.

Some animal items could not be identified due to grinding or digestion. In 2023, some stomachs contained white elastic muscular skins (Figure 2D). They constituted a dense, unraveled lump that occupied the entire stomach. These could be fragments of *Gersemia* sp. or *Sipunculidae* sp., but experts in these groups could not confirm or refute this hypothesis. Since there were numerous skins, they were analyzed separately from the other unidentified animal objects and are called “unidentified skins”.

The microplastics found in the snow crabs’ stomachs were represented by undigested black, blue, red, or transparent microfibers. Their size ranged from 0.5 to 5 mm. A total of 64 microplastic particles were found in 31 crabs’ stomachs.

Temporal changes in the crabs’ diets are only described for the Sill station data since there is no 2018 and 2023 data from the Inner Basin station. The frequency of occurrence (FO) was the highest for detritus and algae (in all years), brittle stars (in 2018), and bivalves (in 2020 and 2023) (Figure 3, Appendix A). The FO of detritus and algae fragments varied slightly between the years and was higher than 75%. A high frequency of brittle stars was observed only in 2018 (64%); in 2020 and 2023, brittle stars were not found in the crabs’ stomachs from the Sill station. As the FO of brittle stars decreased, the frequency of bivalves increased from 25% in 2018 to 71% in 2020 and 88% in 2023. The FO of snow crab fragments was slightly lower in 2018 (20%) than in 2020 (29%) and dropped sharply in 2023 (9%). The frequency of occurrence of foraminifera in 2018 (6%) was noticeably lower than in 2020 (42%) and decreased slightly in 2023 (33%).

The remaining animal components were less common in the crabs’ stomachs. The frequency of polychaetes was 3% in 2018 but increased to 8% in 2020 and 2023. The undefined skins were almost not found in stomachs in 2018 (FO 1%) and 2020 (0%), while in 2023 their FO increased to 7%. The frequency of mysids, shrimps, and other crustacean fragments was small (up to 8%). Gastropods, nematodes, bryozoans, and fish were encountered only occasionally. The frequency of microplastics was similarly low (3–8%) throughout the years.

The average food lump (AFL) (Figure 4, Appendix A) of snow crabs in 2018 was dominated by brittle stars (49%). Detritus (26%), algae fragments (11%), and snow crabs (7%) also play an important role in the crabs’ diet. Small shares in the AFL were occupied by polychaetes (2%), unidentified skins (2%), mysids (2%), and shrimps (1%). The AFL for the 2020 and 2022 data was not calculated due to the small number of full stomachs, but large volume shares of detritus (average value 52% in 2020 and 2022) can be noted in the full stomachs.

The average food lump of 2023 crabs is very different from that in 2018. Brittle stars have completely disappeared from the stomachs. In their absence, the volume fractions of detritus (48%), unidentified skins (23%), polychaetes (9%), and bivalves (10%) have increased. The proportion of algae fragments (3%) decreased, and shrimp and mysids were not included in the AFL. The volume fraction of snow crabs (cannibalism) was 5%.

The Tarverdieva’s coefficient (TC, overall dominance index, Appendix A) of snow crabs’ stomach food items (SFI) in 2018 was 94%, i.e., almost every full stomach contained a certain food item that occupied at least 60% of its volume. The brittle stars (49%) and detritus (25%) had the highest individual dominance index (Appendix A). The individual dominance index of other SFI was less than 10%. There were a few full stomachs in 2020, but each contained a dominant food item (TC 100%). The individual dominance index of detritus was 56%; of brittle stars, it was 33%; and of snow crabs, it was 11%. In 2023, TC was still high (76%), although it was lower than in 2018. The detritus (43%) and unidentified skins (25%) had significantly higher individual dominance indices. Froerman’s coefficient (Fr) shows that in all years of study, the average number of food items in one stomach was higher than 3 (Fr = 3.2 in 2018, 3.8 in 2020, and 3.5 in 2023).

### 3.2. Stable Isotope Analysis

Ninety-one crabs were analyzed using stable isotope analysis (Table 4). The δ^13^C values of all crabs ranged from −21.8 to −19.5‰ and the δ^15^N values from 10.6 to 16.4‰. Isotopic values only slightly positively correlated with the crabs’ size (carapace width CW, mm). The correlation coefficient (Pearson’s product-moment correlation) of δ^15^N values with all crabs’ CW was very weak (r = 0.23) but statistically significant (*p* = 0.03), while the δ^13^C had a stronger correlation (r = 0.34, *p* = 0.04). Nevertheless, when tested separately, male size had a strong positive correlation with the δ^13^C values (r = 0.6, *p* < 0.01) and a weak correlation with δ^15^N (r = 0.34, *p* = 0.03). Females’ size did not correlate with either δ^13^C (r = 0.11, *p* = 0.4) or δ^15^N (r = 0.16, *p* = 0.3) values.

The mode of the posterior ellipse sizes (SEA_B_) is similar to the standard ellipses corrected for small sample sizes (SEA_C_), except for the Inner Basin females from 2020 (Figure 5, Table 4). Therefore, the area of standard ellipses could be used to describe the shape and ascertain the proximity of different crab groups’ isotopic niches, keeping in mind that it can overestimate the size of females from the Inner Basin in 2020. The posterior ellipse sizes (SEA_B_) did not significantly differ between the years, stations, or crabs’ sexes (chi-squared = 27,999, df = 27,999, *p*-value = 0.5, Figure 5). However, the shapes of standard ellipses with 95% data (SEA) have changed from broader in 2018 (along the δ^13^C axis) to taller in 2020 (along the δ^15^N axis) (Figure 6). Even though the shape of SEA of crabs from the Inner Basin station in 2020 seems especially elongated, the statistical analysis did not show significant differences in δ^15^N values between 2018 and 2020, stations, and crabs’ sexes (chi-squared = 11.40, df = 7, *p* = 0.12, Figure 7). On the other hand, the δ^13^C was significantly different (chi-squared = 37.81, df = 7, *p* < 0.01) for some combinations of factors. The δ^13^C values of females from the Sill and males from the Inner Basin station in 2018 differed from females from both stations in 2020 (*p* values < 0.01).

The SEA ellipses incorporating 95% of the data of all crab groups overlapped. The proportion overlap ranged from 0.2 to 0.6 (Figure 7). The smallest overlap was between groups from different years; consequently, the largest overlap was between crabs from the same station and year.

## 4. Discussion

The feeding intensity of crabs can be reflected in the amount of food in their stomachs. The proportion of full stomachs decreased in 2023 (17.5%) compared to 2018 (40.2%). Snow crabs’ feeding activity can alter depending on the time of day [41,53]. Although crabs in this study were collected in different months and times of day, the decrease in the macrobenthos biomass between these years [15] could also lead to decreased consumption.

The diet of snow crabs can vary depending on their size [40,44,52]. The sizes of analyzed crabs increased from 2018 to 2020 and remained within a similar range of an average CW 40–50 mm (Table 1). Male snow crabs can grow to larger sizes, while females are generally smaller, and crabs of different sizes can prefer different depths of habitat [79,80]. Indeed, crabs from the deeper Inner Basin station were larger than those from the Sill station, and most often, males were larger than females. The sizes of dissected crabs coincided with the main size groups of snow crabs in Blagopoluchiya Bay in the corresponding years [15]. By 2023, most crabs were sexually mature (after a terminal molt, according to unpublished data from the Institute of Oceanology), and their carapace width of about 45 mm is smaller than in the natural range and even other parts of the Kara Sea. For example, in the southern bays of Novaya Zemlya, terminal molt occurs in individuals larger than 60 mm [81]. The limited growth of crabs in Blagopoluchiya Bay may be due to low water temperatures since it affects the sizes at which crabs reach sexual maturity [82,83], but it may also be the result of low food supply. The differences in isotopic values and niche sizes between crabs of different sizes cannot be distinguished from the changes in available food items. Since the crabs were large enough to consume macro- and megabenthic species in 2018, the increase in their sizes by 2023 is probably insignificant. It is most likely that crabs of these sizes do not have strong differences in food size preferences. Therefore, differences in their diet are more likely related to food availability.

Identifying prey to the lowest taxonomic level is the primary challenge of visual stomach contents analysis [60]. This is especially difficult since crabs crush their food [56]. We can compensate for this by using available data on the possible prey items in the study area. *Ophiopleura borealis* most likely represented the brittle stars in the stomachs since this species was found in trawl catches at the Sill station in 2018 and the Inner Basin in 2020 [15]. The large chitin fragments are probably the remains of snow crabs since they are the only crab species in Blagopoluchiya Bay [15]. Cannibalism has already been shown for snow crabs both in their natural range [40,41,42,43,84] and in the Kara Sea [53]. However, it remains unknown whether the crabs in the Kara Sea eat their living relatives as active predators or only dead individuals as scavengers.

The organic slur (described as detritus in the results) found in the stomachs may either be part of the consumed bottom sediments (“true” detritus) or heavily digested food. Detritus is usually understood as dead organic matter accumulating in benthic ecosystems as part of bottom sediments. We believe this slur is detritus since mineral components and foraminifera were found in the stomachs, which could only be ingested from bottom sediments. Algae were also found in the stomachs, which do not grow at the depths where the samples were taken and are also part of the detritus. Undoubtedly, some of the organic slur may result from food digestion. Detritus or organic slur was not reported in most snow crabs’ stomachs from their natural range [39,40,41,44,85,86,87,88]. Some studies have noted the presence of detritus in the stomachs, but no description is given, and its proportion in the stomachs is minimal (1% or less by weight in [42,43]. This means that prey animals digested to such a degree do not remain in large quantities in the stomachs. On the other hand, detritus occupied a large volume (up to 100% in some full stomachs) in the Kara Sea snow crabs. Detritus is more often noted in the diet of the snow crabs in the invasive range [50,51,53,54]. However, its detailed description is lacking. The details are only given by Burukovsky et al. [53], describing it as a gray or brown mass that forms flakes in water, which coincides with what we observed.

The visual identification of microplastics (especially transparent) has a fairly large error [89]. This error may further increase for full stomachs with many other stomach content objects. Microplastics in the snow crab stomachs have been described in a small number of studies [47,54,90], whereas in most studies, microplastics have not been observed at all [39,40,53,57,85,87].

The most common method for describing diet based on stomach contents is to determine the frequency of occurrence (FO) of stomach component objects (SCOs) in all stomachs or only in the stomachs with food [59,60]. Stomach food items (SFIs) were found in all dissected snow crab stomachs. Brittle stars, bivalves, detritus, and algae were most common in all stomachs. In empty stomachs (less than 30% full), brittle stars (frequency of occurrence in empty stomachs was 84% in 2018), bivalves (55 and 85% in 2020 and 2023), detritus and algae (more than 80% in all years of the study) also predominated. These SCOs may be digested and absorbed more slowly and remain in the stomachs between the feedings. This would lead to an overestimation of the frequencies of occurrence of these SCOs in all stomachs. However, their quantity in full stomachs was greater than in empty ones. For example, there were up to 10 fragments of bivalve shells in empty stomachs and tens and hundreds of fragments in full stomachs, so it is most likely that they were also ingested in the new feeding act. The frequencies of occurrence of bivalves, detritus, and algae in empty stomachs do not exceed the frequency of occurrence in all stomachs. However, the FO of brittle stars in empty stomachs (84%) is greater than in all stomachs (60%), which may indicate an overestimation of the frequency of occurrence in stomachs relative to the actual frequency of consumption.

Brittle stars were most common in the stomachs from the Sill station (64%) in 2018 and from the Inner Basin station (60%) in 2020. This coincided with brittle stars’ dominance in these benthic communities [15]. By 2020, the number of brittle stars had noticeably decreased [15], and they were no longer found in the crabs’ stomachs and the benthic communities in 2022 and 2023. Predation by snow crabs is the most likely cause of the disappearance of brittle stars in Blagopoluchiya Bay.

In contrast, the FO of bivalves in the crabs’ stomachs increased from 2018 to 2023 (from 25% to 88% at the Sill station). According to benthic surveys [15], the changes in the abundance and biomass of different mollusk species differed. The abundance of *Portlandia arctica* increased in Blagopoluchiya Bay, and *Ennucula tenuis* became one of the dominant species at both stations. Many shell fragments in the stomachs were also attributed to their families. The abundance of other species, *Bathyarca glacialis* and *Astarte* sp., declined markedly in the bay and were not observed in the stomachs. However, it is possible that we simply did not encounter their shell fragments with identifying features.

It is difficult to clarify the identity of the unknown skins in the crabs’ stomachs. The literature describes snow crabs feeding on both sipunculids [50] and *Gersemia* sp. [85]. After the introduction of the snow crab, sipunculid *Golfingia margaritacea* became one of the dominant species in Blagopoluchiya Bay. At the same time, the frequency of occurrence of the unidentified skins increased in 2023 (1% in 2018, 0% in 2020, and 7% in 2023), while other more accessible prey items decreased. Such a coincidence may favor identifying these food items as sipunculids.

After the invasion, the snow crab became one of the dominant megabenthos species [15], and the frequency of occurrence of crab fragments in their stomachs was high in 2018 and 2020. However, the crab fragments FO decreased in 2023. Snow crabs can feed either on juveniles or on molting crabs but not on strong adult individuals [40]. The crabs reached large sizes, and their number decreased between 2018 and 2020, when they could have been dying in large numbers and being easy prey to living specimens. This dying event may be related to the rapid decrease in the biomass of food items in the Blagopoluchiya Bay benthic communities [15].

The frequency of occurrence gives an idea of the frequency of consumption of certain prey items. However, it is important to determine how much of this prey is consumed to understand its role in crabs’ feeding. There are various methods for the quantitative characterization of stomach content objects. Counting the number of fragments in the stomach can lead to overestimating the role of small-sized prey [60]. Weight methods are problematic for decapods due to the vigorous grinding of the prey, although they are still sometimes used [42,43,48]. In this study, we estimated the volume fractions of SCOs and calculated the average food lump using only full stomachs. The average food lump of snow crabs in Blagopoluchiya Bay shows a change in prey animals from brittle stars (mean volume fraction 49%) in 2018 to unidentified skins (23%), polychaetes (9%), and bivalves (10%) in 2023. Also, the proportion of detritus increased significantly in 2023 (48%) compared to 2018 (26%). This is also confirmed by the higher frequency of occurrence of foraminifera in 2023, which was probably ingested with detritus. The volume fractions of crabs (cannibalism) changed little between the years.

Determining the average amount of prey in the stomach (Froerman’s coefficient, Fr) and the frequency of stomachs with a dominant SFI (Tarverdieva’s coefficient, TC) allows us to assess the crab’s feeding strategy. The Froerman’s coefficient (more than 3) and the Tarverdieva’s coefficient (more than 75%) are high for the snow crab in Blagopoluchiya Bay. Regarding their feeding strategy, crabs in the Kara Sea are close to opportunistic predators according to Burukovsky’s classification [60] of decapod feeding strategies. On the one hand, the snow crab’s diet includes many animal food items. It seems that in declining benthic communities of the Kara Sea, the snow crab alternates its predatory feeding with collecting detritus and algae.

The stable isotope analysis results complement the snow crab stomach content analysis. The trophic position (TP) of snow crabs in Blagopoluchiya Bay was assessed in relation to other benthic species in our different work [37]. Crabs are not the top predators in Blagopoluchiya Bay, where the starfish *Urasterias lincki* holds the leading predatory role, followed by several fishes and shrimps. Crabs’ trophic position did not change over time and between the stations (TP 3.0 to 3.4) and remained at the third trophic level (secondary consumer). Similar results were observed in the East Siberian Sea, where the δ^15^N values of the snow crabs are close to those of omnivorous species, which is explained by the predominant feeding on detritivores of low trophic levels (bivalves, sessile polychaetes) or feeding on detritus. In contrast, in the Chukchi Sea, the snow crabs’ δ^15^N values are higher, presumably due to the greater role of cannibalism [84]. In the Bering Sea, the δ^15^N values of the snow crabs are on par with polychaetes and brittle stars [91]. Brittle stars in Blagopoluchiya Bay are deposit feeders with low TP (2.8) at a trophic level of the primary consumer but closer to the secondary consumer [37]. The trophic positions of other crab’s preferred prey organisms (mollusks, polychaetes) have a broad range. However, the TPs of the most abundant species (including sipunculids) are close to the ophiuroids [37].

Overall, the isotopic niches of crabs of different sexes and from different stations were very similar between the years (strongly overlapped), suggesting very similar feeding habits throughout the study. The variety of food (food spectrum) does not quantitatively change across time, sex, and sampled stations (similar sizes of isotopic niches, Figure 5). However, there are differences in the sources of this variety, reflected in the changes in the shape of these niches (Figure 6). Crabs drew nutrition from broader organic carbon sources in 2018 than in later years. The broader (along the δ^13^C axis) range of isotopic niches in crabs from 2018 could be the result of a larger variety of prey from different feeding modes. The nitrogen isotope values and niche ranges (along the δ^15^N axis) remained similar, supporting the conclusions drawn from trophic position calculations that the changes in prey items observed in this study occurred within a similar trophic level range.

Our results align with the few available stomach content studies from the Kara Sea. Earlier studies show that detritus was not found in the stomachs of snow crabs in 2014–2016 in the southeastern part of the Kara Sea, and different groups of invertebrates prevailed (in terms of frequency of occurrence and mass fraction): bivalves, polychaetes, crustaceans, and brittle stars [48,49]. Burukovsky et al. [53] observed lower frequencies of occurrence of SCOs in the stomachs of crabs in Blagopoluchiya Bay in 2018. However, our study did not take into account the contents of obviously empty stomachs, and Burukovsky’s work provides combined data from the Inner Basin and the Sill stations while we look at them separately. Nevertheless, the most common SCOs are the same, and the descriptions of the average food lump and the overall dominance indices are consistent.

In another part of snow crabs’ invasive range, the Barents Sea, most of the macrozoobenthos groups have been described in the diet of the snow crab. In different parts of the region, crustaceans, polychaetes, and bivalves predominate in the stomachs of crabs, as well as brittle stars, holothurians, fish, etc. [45,46,47,49,50,51,52]. Detritus is less common in the stomach, but its role in nutrition has not been assessed [50,51]. In the Pechora Sea (southeastern Barents Sea), the contents of snow crabs’ stomachs were dominated by bivalves (37% of AFL) and detritus (36%) [54].

In its natural range, the snow crab is described as a predatory or omnivorous species, but animal food predominates in its stomachs: polychaetes, bivalves, decapods, brittle stars, or other groups of invertebrates, which are usually most numerous in the area where the crabs are collected [39,40,41,44,85,86,87,88]. Detritus is rarely found in the diet but accounts for a small proportion of the stomachs [42,43]. Only in the East Siberian Sea does the low δ^15^N value of the snow crab suggest an important role of detritus in nutrition [84]; however, there are no stomach content studies to support this.

In 2018, the most common macro- and megabenthic species were similar to those found before the invasion [15]. The most drastic changes occurred between 2018 and 2020. This is also reflected in the changes in food items in crabs’ stomachs. The changes in the frequency of occurrence of the main animal SFI of the snow crab (brittle stars, bivalves, crabs) coincide with changes in the macro- and megabenthos communities of Blagopoluchiya Bay. Thus, the diet of the snow crab in Blagopoluchiya Bay of the Kara Sea in 2018 was similar to that of crabs in the Barents Sea, where animal food predominates while detritus is also present in its diet. However, in 2023, detritus occupies almost half of the average food lump in Blagopoluchiya Bay, which significantly distinguishes the diet of the crabs in the Kara Sea from other regions. Probably, in the conditions of depleted benthic communities in Blagopoluchiya, the snow crab is forced to use all available food sources.

## 5. Conclusions

By 2022, the most common benthic species found in the bay were the snow crabs, predatory seastars, and less attractive food items such as sipunculids, anemones, and sponges. Previously commonly found ophiuroids have completely disappeared. These changes were reflected in the food items found in the snow crabs’ (*Chionoecetes opilio*) diet in Blagopoluchiya Bay after 2018, where the proportion of bivalves and detritus has substantially increased in the crabs’ diet.

The changes in prey did not lead to changes in crabs’ trophic status. The crabs’ isotopic niche remained similar throughout the years. The changes in crabs’ diet have occurred within the same trophic level.

After the change in crabs’ diet and changes in the Blagopoluchiya Bay benthic community, the crab population has decreased. We expect a further decrease in the number of crabs in Blagopoluchiya Bay, with the possible full of partial restoration of the benthic community of the bay and a repeat of the cycle after new settlements of young crabs in the bay.

Further monitoring of these population fluctuations could reveal the ability of the arctic fjord benthic communities to restore and adapt to the new invader. In addition, new settlements of the crab in the bay can shed light on the factors that allowed it to settle in 2014 and have restricted new settlements since then. Such information could help in assess the possibility of establishing a commercial population of snow crabs in the Kara Sea.

## Figures and Tables

**Figure 1 biology-13-00781-f001:**
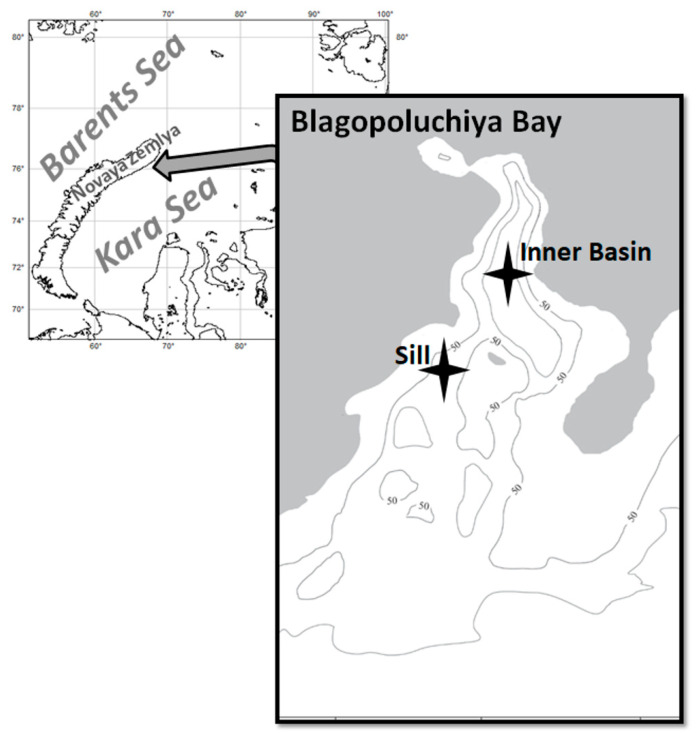
A map of Blagopoluchiya Bay with sampling sites marked by stars.

**Figure 2 biology-13-00781-f002:**
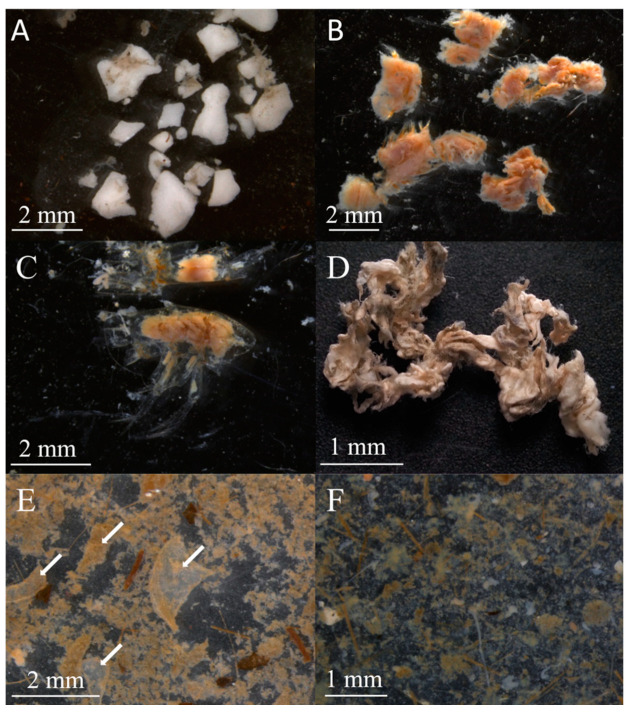
Photographs of some stomach content objects of *Chionoecetes opilio* from Blagopoluchiya Bay sampled in 2018 and 2023. (**A**) Skeleton elements of Ophiuroidea; (**B**) fragments of polychaete group Errantia; (**C**) thoracic segments of a crustacean; (**D**) unidentified skin; (**E**) carapace fragments of *C. opilio* (highlighted by arrows); (**F**) detritus and algae fragments.

**Figure 3 biology-13-00781-f003:**
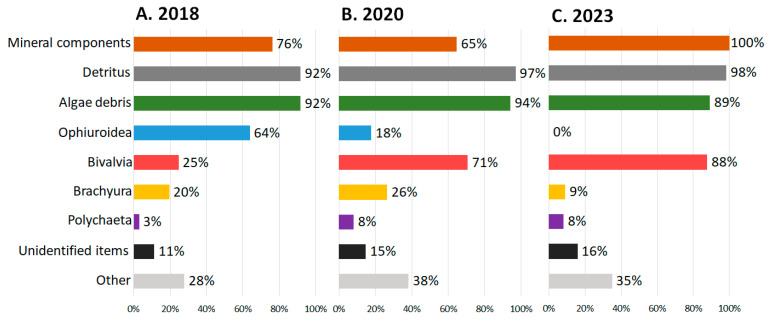
The frequency of occurrence of stomach content objects in *Chionoecetes opilio*, collected from the Blagopoluchiya Bay Sill station in (**A**) 2018, (**B**) 2020, and (**C**) 2023.

**Figure 4 biology-13-00781-f004:**
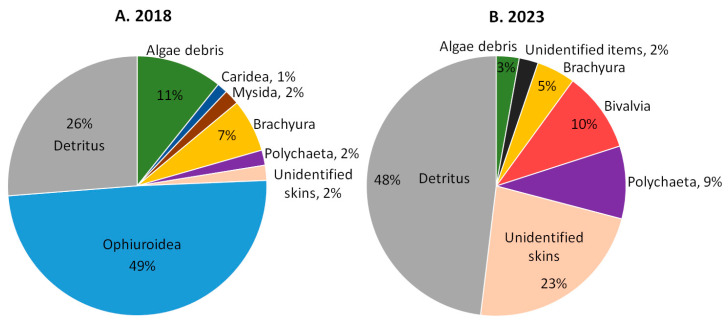
The volume proportions of stomach content objects in the average food lump of *Chionoecetes opilio* from the Sill station of Blagopoluchiya Bay collected in (**A**) 2018 and (**B**) 2023.

**Figure 5 biology-13-00781-f005:**
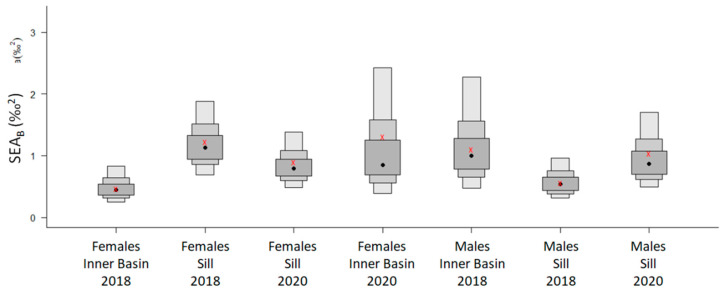
Isotopic niche density boxplots of the Bayesian estimates of Standard Ellipse Area (SEA_B_) of female and male *Chionoecetes opilio* sampled from the Inner Basin and Sill stations in Blagopoluchiya Bay in 2018 and 2020. Boxes represent 50, 75, and 95% credible intervals in decreasing order of size, with a black circle indicating SEA_B_ mode and a red cross representing the Standard Ellipse Area corrected for small sample size (SEA_C_).

**Figure 6 biology-13-00781-f006:**
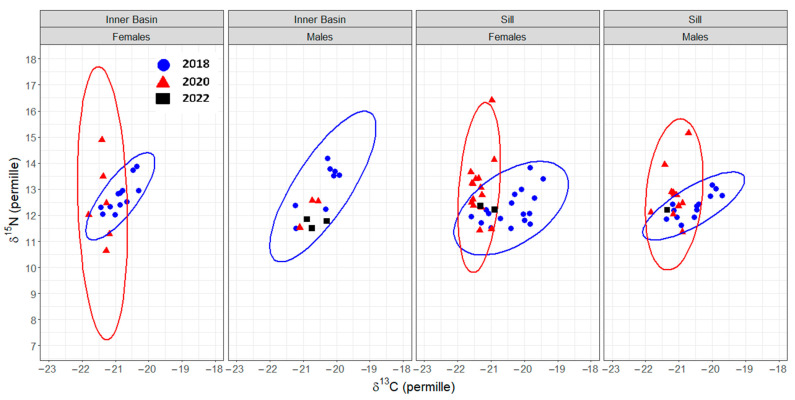
Isotopic niche, approximated by ellipses that incorporate 95% of the data (SEA), of female and male *Chionoecetes opilio* from the Inner Basin and Sill stations of Blagopoluchiya Bay sampled in 2018 (blue) and 2020 (red). Data from 2022 (black) and males from the Inner Basin in 2020 are added as data points due to the small sample size.

**Figure 7 biology-13-00781-f007:**
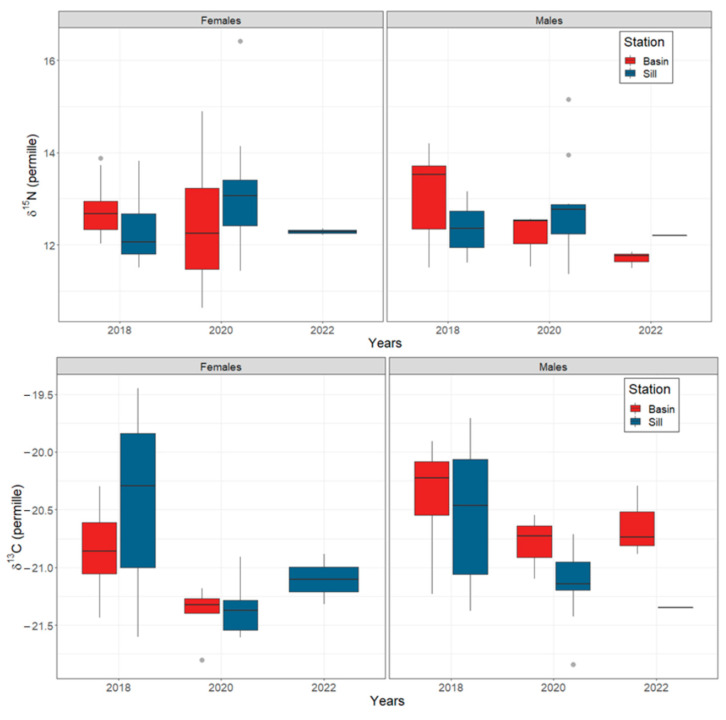
Boxplots of δ^15^N and δ ^13^C (in ‰ units) of female and male *Chionoecetes opilio* sampled from the Inner Basin (red boxes) and Sill (blue boxes) stations in Blagopoluchiya Bay in 2018, 2020, and 2022. Inbox lines indicate the mode, boxes indicate upper and lower quartiles, and whiskers indicate 1.5 quartile ranges. Grey circles are outliers.

**Table 1 biology-13-00781-t001:** The mode, mean ± standard deviations, and size range (minimum to maximum carapace width, mm) values of a number (*n*) of female and male snow crabs, *Chionoecetes opilio*, sampled from the Inner Basin and Sill stations in Blagopoluchiya Bay in 2018, 2020, 2022, and 2023. The total female values for 2020, 2022, and 2023 are presented, which include values of ovigerous females (Fov) written in brackets in *italic* font.

Sex	Station	Year	*n*	Mode	Mean ± SD	Range
Females	Inner Basin	2018	12	31	38.7 ± 12.1	23 to 59
	Sill	2018	63	30	30.6 ± 8.2	14 to 72.8
	Inner Basin	2020	6	38.5	42 ± 2.2	38.5 to 44.5
	*(Including Fov)*		*(1)*	*(44.5)*		
	Sill	2020	15	35	35.6 ± 5.4	28 to 45.5
	*(Including Fov)*		*(7)*	*(36)*	*(40.5 ± 3.8)*	*(36 to 46)*
	Sill	2022	3	43.3	41.3 ± 3.5	37.2 to 43.3
	*(Including Fov)*		*(1)*	*(43.3)*		
	Sill	2023	91	38.3	41.2 ± 3.4	34.4 to 53.3
	*(Including Fov)*		*(89)*	*(38.3)*	*(41.2 ± 3.4)*	*(34.4 to 53.3)*
Males	Inner Basin	2018	8	27	45.4 ± 13.9	27 to 66
	Sill	2018	64	28	29.4 ± 6.8	16 to 56
	Inner Basin	2020	5	32.5	48.1 ± 9.9	32.5 to 59.5
	Sill	2020	10	33	33.1 ± 3.7	28.5 to 41
	Inner Basin	2022	5	44.1	49.4 ± 6	44.1 to 59.2
	Sill	2022	1	40.1		
	Sill	2023	29	43.8	50.1 ± 6.3	40.1 to 66.9

**Table 2 biology-13-00781-t002:** The number of sampled and analyzed snow crabs, *Chionoecetes opilio*, for stomach content analysis, from Blagopoluchiya Bay in 2018, 2020, 2022, and 2023.

	2018	2020	2022	2023
Total number of sampled *C. opilio*	132	35	9	120
Total number of analyzed stomachs	97	34	9	120
Number of analyzed stomachs from the Sill st.	97	24	4	120
Number of analyzed stomachs from the Inner Basin st.	0	10	5	0
Total number of full stomachs	53	9	4	21
Proportion of full stomachs, %	40.2	25.7	44.4	17.5

**Table 3 biology-13-00781-t003:** List of stomach content objects found in *Chionoecetes opilio*, collected from Blagopoluchiya Bay in 2018, 2020, 2022, and 2023.

Stomach Content Objects	Taxa	Group
Annelida gen. sp.	Annelida	Animal components
*Spiochaetopterus* sp.	Annelida	Animal components
Nephtyidae gen. sp.	Annelida	Animal components
Bivalvia gen. sp.	Bivalvia	Animal components
Nuculanida gen. sp.	Bivalvia	Animal components
*Nuculana* sp.	Bivalvia	Animal components
Nuculida gen. sp.	Bivalvia	Animal components
*Ennucula* sp.	Bivalvia	Animal components
*Yoldiella* sp.	Bivalvia	Animal components
*Portlandia* sp.	Bivalvia	Animal components
*Ciliatocardium* sp.	Bivalvia	Animal components
Gastropoda gen. sp.	Gastropoda	Animal components
Crustacea gen. sp.	Crustacea	Animal components
Mysida gen. sp.	Crustacea	Animal components
Caridea gen. sp.	Crustacea	Animal components
*Eualus* sp.	Crustacea	Animal components
*Pandalus* sp.	Crustacea	Animal components
*Chionoecetes opilio*	Crustacea	Animal components
Ophiuroidea gen. sp.	Ophiuroidea	Animal components
Bryozoa gen. sp.	Bryozoa	Animal components
*Alcyonidium* sp.	Bryozoa	Animal components
Nematoda gen. sp.	Nematoda	Animal components
Foraminifera gen. sp.	Foraminifera	Animal components
*Haplophragmoides* sp.	Foraminifera	Animal components
*Dentalina* sp.	Foraminifera	Animal components
*Lenticulina* sp.	Foraminifera	Animal components
*Elphidium* sp.	Foraminifera	Animal components
*Triloculina* sp.	Foraminifera	Animal components
*Cornuspira* sp.	Foraminifera	Animal components
*Cibicides* sp.	Foraminifera	Animal components
Fish	-	Animal components
Unidentified items	-	Animal components
Algae debris	-	Algae debris
*Chordaria* sp.	Ochrophyta	Algae debris
*Desmarestia* sp.	Ochrophyta	Algae debris
*Ptilota* sp.	Rhodophyta	Algae debris
*Ectocarpus* sp.	Ochrophyta	Algae debris
*Dictyosiphon* sp.	Ochrophyta	Algae debris
Diatoms	Ochrophyta	Algae debris
Detritus	-	Detritus
Sand particles	-	Mineral components
Stones	-	Mineral components
Silt	-	Mineral components
Fat globules	-	-
Microplastic	-	Microplastic

**Table 4 biology-13-00781-t004:** The number (*n*) of crabs, their mean δ^13^C and δ^15^N values ± standard deviation, standard ellipse area corrected for small sample size (SEA_C_ ‰^2^), and the mean and mode of posterior standard ellipse areas (SEA_B_ ‰^2^) of female and male *Chionoecetes opilio* crabs from the Inner Basin and Sill stations collected in 2018, 2020, and 2022.

	Station	Year	*n*	δ^13^C ‰	δ^15^N ‰	SEAc ‰^2^	Mean SEA_B_ ‰^2^	Mode SEA_B_ ‰^2^
Females	Inner Basin	2018	12	−20.9 ± 0.4	12.7 ± 0.6	0.5	0.5	0.5
	Sill	2018	17	−20.4 ± 0.6	12.3 ± 0.7	1.3	1.3	1.1
	Inner Basin	2020	6	−21.4 ± 0.2	12.5 ± 1.5	1.3	1.3	0.9
	Sill	2020	15	−21.4 ± 0.2	13.1 ± 1.2	0.9	0.9	0.8
	Sill	2022	2	−21.1 ± 0.3	12.3 ± 0.1			
Males	Inner Basin	2018	8	−20.4 ± 0.5	13.1 ± 0.9	1.1	1.3	1
	Sill	2018	13	−20.6 ± 0.6	12.4 ± 0.5	0.6	0.6	0.5
	Inner Basin	2020	3	−20.8 ± 0.3	12.2 ± 0.6			
	Sill	2020	11	−21.1 ± 0.3	12.8 ± 1.0	1	1	0.9
	Inner Basin	2022	3	−20.6 ± 0.3	11.7 ± 0.2			
	Sill	2022	1	−21.4	12.2			

## Data Availability

Raw data of stomach content analysis will be provided in the Appendix A of this article.

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
