# Peer review of "Changes in the Diet of an Invasive Predatory Crab, Chionoecetes opilio, in the Degrading Benthic Community of an Arctic Fjord"

_biology, 2024, doi:10.3390/biology13100781_

Round 1
Reviewer 1 Report
Comments and Suggestions for Authors
This is an innovative and practical study that provides new perspectives for understanding the impact of invasive alien species on ecosystems through detailed research design and scientific data analysis. Thus, the manuscript can be considered for publication after appropriate revision and improvement.
a. Why Arctic fjords were chosen as a study site? The authors need to provide more background on this.
b. In the discussion of results section, comparative analyses of other similar cases could be added to further explore the generalisability and limitations of the findings of this study.
c. A brief description of the method of statistical analysis of the data could also be given to increase the reliability of the results.
d. In the conclusion section, in addition to summarising the results of the study, the direction of future research and the possible practical application value can be clearly presented.
Author Response
Dear Reviewer,
Thank you very much for reading and analyzing our manuscript, and for your helpful comments. Following are our replies to the specific comments.
- Why Arctic fjords were chosen as a study site? The authors need to provide more background on this.
Reply: In the introduction section we discuss why we have chosen Blagopoluchiya Bay as a model study area. The following reasons are given
- The Arctic Kara Sea is interesting because (line 61) “Indeed, due to adverse climate conditions, until recently, this sea experienced 61 low human impact levels, which changed after the sea experienced a slowdown in the formation of sea ice in autumn and its earlier destruction in spring-early summer [26].”
- Fjords are vulnerable and have lower functional redundancy of macrofauna and are therefore simpler model communities to study the changes(line 71) “In particular, fjords are highly vulnerable to anthropogenic and climate-related impacts due to the lower functional redundancy of macrofauna compared to the open sea [36].”
- This particular bay was well studied before and during the invasion, so we have good data. It is hard to predict invasions, so unfortunately, not many places have enough data prior to the invasion. (line 76) “The benthic ecosystem of this bay was strongly influenced by the invasion of the snow carbs, resulting in changes in taxonomic structure, abundance, and biomass of native species [15]. This bay was studied 78 prior to and during the invasion [15,27,37], which makes Blagopoluchiya Bay a unique model fjord to study the invasion of a predator in an otherwise pristine ecosystem.
We can’t think of better reasons for choosing this model area.
- In the discussion of results section, comparative analyses of other similar cases could be added to further explore the generalisability and limitations of the findings of this study.
Reply: We are not certain what kind of “comparative analyses” are implied. However in the introduction we have discussed the use of stomach content and stable isotope analysis and their limitations regarding crab’s feeding habits (lines 101-135). In the discussion section we have compared our study to numerous studies of snow crabs feeding habits from native and invasive ranges (570-593) that used both stomach contents and stable isotope analyses. We have compared our results with these studies and discussed both methodological and biological reasons for the observed differences.
- A brief description of the method of statistical analysis of the data could also be given to increase the reliability of the results. .
Reply: The description of statistical analyses is given in “2.4. Data analysis” subsection of the M&M section of the manuscript. The SIBER analysis is described in detail with references on the literature that explains this widely used method in stable isotope studies. The non-parametric Kruskal-Wallis test was described giving dependent variables (CW, δ13C, and δ15N) and factors (sexes, sampling stations, and years). The post-hoc test was named Dunn multiple comparison test and its p values adjustment method (Holm method). These are standard, widely used methods and describing them in more detail seems redundant. If we missed some particular questionable feature of the analyses, please let us know which one.
- In the conclusion section, in addition to summarising the results of the study, the direction of future research and the possible practical application value can be clearly presented.
Reply: we have added “Further monitoring of these population fluctuations could reveal the ability of the arctic fjord benthic communities to restore and adapt to the new invader. In addition, new settlements of the crab in the bay can shed light on the factors that have allowed it to settle in 2014 and restricted new settlements since then. Such information can be helpful to assess the possibility of the establishment of commercial population of snow crabs in the Kara Sea.”
Reviewer 2 Report
Comments and Suggestions for Authors
Manuscript ID: biology-3195684
Title: Changes in the diet of an invasive predatory crab, Chionoecetes opilio, in the degrading, invaded benthic community of an Arctic fjord.
Review
Authors aimed to analyze changes in the feeding habits of the invasive snow crabs, Chionoecetes opilio, in the degrading benthic ecosystem of an Arctic fjord, Blagopoluchiya Bay, under the influence of this invader. As a method they used use stomach content and stable isotope analyses.
As the crab is, as authors say, the only factor of degradation in the investigated ecosystem, study poses scientific interest as an example of changes in aquatic environment.
I have no negative comments as for the methods and used statistics (they used SIBER for stable isotope analysis, which is de facto in such investigations), and conclusions are supported by results. Language is fully understandable; most possibly, some writing program was used instead of native speaker, so no comments as for the text.
However, before acceptance there is a need of revision. Please have my comments below.
General comment 1: please stick to Template in revision.
General Comment 2: Do not repeat data from Table in Figures. Comments below.
General Comment 3: aim is to reveal changes; therefore, analyses should be related mainly to this aspect.
General Comment 4: format References as required by Template.
Critical comments
1. Table 1 and Figure 2 duplicate each other and use the same data. Remove one of these.
2. Table 2 is not needed here in the text, you may use asterisks or other notation to indicate significant differences in either Table 1 or Figure 2, which one you choose to not to remove.
3. Figure 4 duplicates Table 5 (columns 2 to 4). It is not allowable, make revision or remove
4. Figure 5 also duplicates Table 5, remove figure
5. Table 5 uses proportions. I miss statistics, if between years these proportions differ significantly. G-test might be used, or z-test for proportions
6. As the aim was to analyze changes, Figure 7 should be separated into 4 part: Basin, males; Basin, females; Sill males, Sill females – ellipses correspond to year. Then it will be clear visually, in the current form it is not readable.
7. Line 626: Supplement should be presented as required by MDPI, Table S1. Table caption, and referred accordingly in the text. On the positive note, I appreciate giving raw material available as Supplement.
Specific comments:
Line 3: remove “, invaded” from the Title
Lines 118–124: this belong to Discussion, or Material and Methods.
Lines 179–180: maybe the opposite? Why empty stomachs were preserved?
Line 171: define abbreviation at first use only
Line 177: define abbreviation at first use only
Line 196: use en-dash for ranges, throughout the text, e.g., n = 4–8
Line 198: correct abbreviation for standard deviation is SD, not sd
Line 226: mistype
Line 397: mistype? (chi-squared = 27999, df = 27999
Lines 419–420: I understand authors want to say, that opportunistic collection of samples remove bias due to feeding patterns, but please find better wording if I am right.
Line 475: abbreviations already defined many times
Line 560: “all groups of crabs” – which groups? Sex, size or else?
Line 646: remove Appendix, as you suppose to give Supplemental Table S1
Acknowledgments: you use different way of surname and initials
Conflict of interests: check Template, and add the role of funders in …
Comments on the Quality of English LanguageLines 419–420: I understand authors want to say, that opportunistic collection of samples remove bias due to feeding patterns, but please find better wording if I am right.
Author Response
Dear Reviewer,
Thank you very much for reading and analyzing our manuscript, and for your helpful comments. Following are our replies to the specific comments.
In relation to the “most possibly, some writing program was used instead of native speaker,” comment, we would like to express our opinion.
The second author of this paper is bilingual, graduated from English school and universities, and her “native” Russian language skills are appalling. We believe that the place of birth or passport do not necessarily make “native” speakers more proficient in language skills. Many “non-native” speakers are considered great writers in languages they have studied in their life (consider Nabokov). In addition, “native” speaker professional writers (such as journalists) have several editors correcting their texts before the publication. Even great scientists are often poor writers (consider original Darwin texts). Therefore, we find it odd that in the modern “politically correct” world, where English is the most studied language, the “nativeness” of the author is used as criteria of their proficiency.
General comment 1: please stick to Template in revision. – Reply: in the instructions for submission it is stated that the formatting of the manuscript and references is finalized after the acceptance.
“ï‚· Your references may be in any style, provided that you use the consistent formatting throughout. It is essential to include author(s) name(s), journal or book title, article or chapter title (where required), year of publication, volume and issue (where appropriate) and pagination. DOI numbers (Digital Object Identifier) are not mandatory but highly encouraged. The bibliography software package EndNote, Zotero, Mendeley, Reference Manager are recommended.
ï‚· When your manuscript reaches the revision stage, you will be requested to format the manuscript according to the journal guidelines.”
However, we will try to do as much as we can now.
General Comment 2: Do not repeat data from Table in Figures. Comments below. – Reply is given below, in critical comments.
General Comment 3: aim is to reveal changes; therefore, analyses should be related mainly to this aspect. – Reply: We are not quite certain what is implied in this comment. The changes are revealed using stomach content analysis, while the stable isotope analysis has shown that they were within the same trophic level. This does not imply that there were no changes. We will attempt to define these results more clearly in the conclusion section.
General Comment 4: format References as required by Template. – Reply: in the instructions for submission it is stated that the formatting of the manuscript and references is finalized after the acceptance. We will be happy to do so when the editor instructs us to.
Critical comments
- Table 1 and Figure 2 duplicate each other and use the same data. Remove one of these.
- Table 2 is not needed here in the text, you may use asterisks or other notation to indicate significant differences in either Table 1 or Figure 2, which one you choose to not to remove.
- Figure 4 duplicates Table 5 (columns 2 to 4). It is not allowable, make revision or remove
- Figure 5 also duplicates Table 5, remove figure
Reply to comments 1, 2, 3 and 4 – Indeed, the figures somewhat duplicate the information in the tables, however they make this information much more accessible and the differences visually apparent. The information in the tables is much more detailed than in the figures, and not fully reflected in the figures. The manuscript is not too long (not over 40 pages) and the tables and figures are not too numerous. The authors would like to keep both, the figures and the tables. However, if there is a strict rule against this, we will abide.
- Table 5 uses proportions. I miss statistics, if between years these proportions differ significantly. G-test might be used, or z-test for proportions. Reply: Thank you for this suggestion. We have used two-proportion z-test to compare proportions from 2020 and 2022 with 2018, which should reflect the biggest differences. This information is added to table 5, and described in the methods section.
- As the aim was to analyze changes, Figure 7 should be separated into 4 part: Basin, males; Basin, females; Sill males, Sill females – ellipses correspond to year. Then it will be clear visually, in the current form it is not readable. – Reply: Thank you for this very helpful suggestion. We have done as you recommended.
- Line 626: Supplement should be presented as required by MDPI, Table S1. Table caption, and referred accordingly in the text. On the positive note, I appreciate giving raw material available as Supplement. – Reply: The information has been added.
Specific comments:
Line 3: remove “, invaded” from the Title – Reply: Deleted
Lines 118–124: this belong to Discussion, or Material and Methods. – Reply: We are discussing the different methods used to study crab’s feeding habits. There are many coefficients and ways to describe stomach contents. We believe that some introduction and reasoning for choosing these methods is appropriate in the Introduction section of the manuscript.
Lines 179–180: maybe the opposite? Why empty stomachs were preserved? – Reply: The coefficients and average food lump was calculated only for full stomachs because it is volumetric criteria and therefore can lead to errors when used for different fullness of stomachs (line 174) “Using this indicator only for full stomachs allows us to avoid errors associated with different digestion rates of SCOs.” We still looked at some empty stomachs to reveal this possible difference in the digestion (Line 254) “However, some empty stomachs were analyzed and revealed that they, too, contain a small portion of items.”, which is discussed in the Discussion section (Lines 477-484) “In empty stomachs (less than 30% full), brittle stars (frequency of occurrence in empty stomachs was 84% in 2018), bivalves (55 and 85% in 2020 and 2023), detritus and algae (more than 80% in all years of the study) also predominated. These SCOs may be digested and absorbed more slowly and remain in the stomachs between the feedings. This would lead to an overestimation of the frequencies of occurrence of these SCOs in all stomachs. However, their quantity in full stomachs is greater than in empty ones. For example, there were up to 10 fragments of bivalve shells in empty stomachs and tens and hundreds of fragments in full stomachs, so it is most likely that they were also ingested in the new feeding act.”
Line 171: define abbreviation at first use only – Reply: deleted definition of the abbreviation
Line 177: define abbreviation at first use only – Reply: Tarverdieva’s coefficient is defined in this section of the manuscript for the first time. We believe that we use many abbreviations and it would be hard for the readers to remember them all, so at least one definition should be supplied in each section.
Line 196: use en-dash for ranges, throughout the text, e.g., n = 4–8. – Reply: Changed everywhere.
Line 198: correct abbreviation for standard deviation is SD, not sd. Reply: Changed
Line 226: mistype Reply: We are sorry, we can’t spot it. Probably we are too used to our text to spot some errors.
Line 397: mistype? (chi-squared = 27999, df = 27999). – Reply: No, this is the actual result we got in R. There were 28000 SEAB measurements from the SIBER analysis, and it is probably a coincidence that the chi-squared is the same.
Lines 419–420: I understand authors want to say, that opportunistic collection of samples remove bias due to feeding patterns, but please find better wording if I am right.
Reply: The collection time was random due to different timing of the expeditions and different workload in the Bay. Unfortunately it was not intentional, and instead of removing bias, it can hinder the changes between the years. We have re-written this paragraph, hopefully clarifying this argument.
“The feeding intensity of crabs can be reflected in the amount of food in their stomachs. The proportion of full stomachs decreased in 2023 (17.5%) compared to 2018 (40.2%). Snow crabs’ feeding activity can alter depending on the time of [41,53]. Although, crabs in this study were collected in different months and times of day, the decrease in the macrobenthos biomass between these years [15] could also lead to de-creased consumption.”
Line 475: abbreviations already defined many times only – Reply: These abbreviations were not defined in this section of the manuscript. We believe that we use many abbreviations and it would be hard for the readers to remember them all, so at least one definition should be supplied in each section.
Line 560: “all groups of crabs” – which groups? Sex, size or else? – Reply: Clarified by additional information “Overall, the isotopic niches of crabs of different sexes and from different stations were very similar between the years (strongly overlapped)…”
Line 646: remove Appendix, as you suppose to give Supplemental Table S1 – Reply: Deleted Appendix, added information in the Supplement material section.
Acknowledgments: you use different way of surname and initials – Reply: Corrected
Conflict of interests: check Template, and add the role of funders in …
Reply: We have copy pasted the conflict of interest declaration from the website
https://www.mdpi.com/journal/biology/instructions
“If no conflicts exist, the authors should state:
Conflicts of Interest: The authors declare no conflicts of interest.”
We have added the following statement about the funding:
“Both authors have received research grants from Russian Science Foundation, that does not restrict authors’ bility to analyze and interpret the data and to prepare and publish manuscripts independently when and where they choose.”
Comments on the Quality of English Language
Lines 419–420: I understand authors want to say, that opportunistic collection of samples remove bias due to feeding patterns, but please find better wording if I am right. – Reply: This is the same as a comment above. See the reply supplied there.
Reviewer 3 Report
Comments and Suggestions for Authors
The authors present an interesting study on the changes in the diet of an invasive predatory crab following the degradation of an Arctic fjord caused by the invasion of the crab. The study was conducted considering two approaches: stomach content analysis and isotopic analysis. I recommend that the sections on stomach content analysis in Results and Discussion be shortened without losing essential information.
My main comments on the manuscript are the following:
- Abstract
After reading the abstract, I assumed that an analysis was conducted to determine the trophic position of the species. However, after reading the manuscript, it is not clear to me how this study was carried out. There is no mention of it in either the Materials and Methods or the Results sections. This part needs to be clarified (and included in M&M and Results), as it is important as part of the study's objectives. Please, provide as much information as possible on this issue.
L. 208: “As THE ABUNDANCE OF the most common prey items diminished”.
- Introduction
I really liked the introduction. It's very informative and well-organized.
- Material and Methods
L. 142. Please, use italics for the Latin name of the species.
L. 175-176. Review the paragraph. I suggest: “Using this indicator only for full stomachs ALLOWED us to avoid errors associated with different digestion rates of SCOs”.
L. 201: The authors indicate that the isotopic niche parameters were calculated for crabs of different sexes. Are data on size classes also available? It would improve the trophic information on the species and potential changes due to growth.
- Results
L. 220-222. In the following paragraph “…..reflects the most numerous size group sampled in the bay that year”. Which year?
L. 226: "Table1" must be changed to "Table 1".
L. 257: The text “Further, only the results of analyzed stomachs are reported”, refers to full stomachs?
L. 266: Use “microplasticS”.
L. 361-362 (and Table 6). Are means for 13C and 15N significantly different?
L. 406-409: This is a crucial aspect of the study. To fully illustrate this point, a figure depicting trophic ellipses for various global groups (across years/seasons) should be included, akin to Figure 7.
L. 402-403: “On the other hand, the δ13C was significantly different (chi-squared = 37.81, df = 7, p< 0.01) for some combinations of factors”. Please, explain. What factors would be more interesting to highlight?
Table 1: I suggest “Range” instead of “Mean to max” (last column and table legend).
I suggest that Table 5 and Figure 6 be transferred to Supplementary Material.
Fig. 7 is difficult to analyse. I suggest increasing its width.
- Discussion
L. 419: The sentence “Crabs were collected in different months and times of day” should be considered in M&M.
L. 423: The authors claim that “The sizes of analyzed crabs increased from 2018 to 2020 and remained within a similar range”. What range are you referring to?
L. 430: “…most crabs were sexually mature”. Over which months did maturation take place?
L. 438-439: I do not understand the following statement: “The sizes between 2018 and 2023 are insignificant in relation to the sizes of common macrobenthic species”.
L. 469: What kind of error are you referring to?
L. 483: “However, their quantity in full stomachs WAS greater 483 than in empty ones”.
L. 492-493. In relation to the text “This coincided with brittle stars’ dominance in these benthic communities”, understanding the sampling periods for each annual cycle is crucial. When did brittle stars’dominance occur?
L. 522: Use “in crabs’ feeding” instead of “in crabs’ nutrition”.
L. 547-549. “Crabs’ trophic position did not change over time and between the stations (TP 3.0 to 3.4), and remained at the third trophic level (secondary consumer)”. This has not been considered in M&M or Results. Also, what organism was the baseline reference in the analysis? Also, given the potential for temporal and spatial heterogeneity in baseline isotopic values, the authors should carefully consider the implications for estimating changes in a species' trophic level. This variability can introduce uncertainty into trophic position estimates.
L. 549-552. Provide references for this statement.
L. 596-597: A deeper focus on the dramatic shifts in macro- and megafauna populations between 2018 and 2020 is needed. What changes were observed and what were the underlying causes?
- Conclusions
L. 614: “The ophiuroids have disappeared from the bay” is already stated in L. 610-611.
L. 609-617: Please, rearrange and condense the content of this section of the conclusions.
Author Response
Dear Reviewer,
Thank you very much for reading and analyzing our manuscript, and for your helpful comments. Following are our replies to the specific comments.
- Abstract
After reading the abstract, I assumed that an analysis was conducted to determine the trophic position of the species. However, after reading the manuscript, it is not clear to me how this study was carried out. There is no mention of it in either the Materials and Methods or the Results sections. This part needs to be clarified (and included in M&M and Results), as it is important as part of the study's objectives. Please, provide as much information as possible on this issue.
Reply: Indeed, the trophic position is not the result of this study, as it requires a large and lengthy analysis, as you have suggested in the comment below “what organism was the baseline reference in the analysis? Also, given the potential for temporal and spatial heterogeneity in baseline isotopic values”. We have prepared and submitted a different paper dedicated to the trophic position of the crab and most common benthic organisms of the bay (Zalota et al. “Trophic position stability of benthic organisms in a changing food web of an Arctic fjord under the pressure of an invasive predatory snow crab, Chionoecetes opilio.”, Biology, submitted). Hopefully these papers will be published within similar timeframe and complement each other. Putting both of these studies in one paper would result in a lengthy and complicated text. We have changed “trophic position” to “trophic niche” in the abstract section, since this is the result of this study, and attempted to clarify trophic position calculation references in the discussion section (see reply below).
- 208: “As THE ABUNDANCE OF the most common prey items diminished”. – Reply: We suppose that this refers to line 28. Corrected.
- Introduction
I really liked the introduction. It's very informative and well-organized.
- Material and Methods
- 142. Please, use italics for the Latin name of the species. – Reply: Corrected
- 175-176. Review the paragraph. I suggest: “Using this indicator only for full stomachs ALLOWED us to avoid errors associated with different digestion rates of SCOs”. – Reply: Corrected
- 201: The authors indicate that the isotopic niche parameters were calculated for crabs of different sexes. Are data on size classes also available? It would improve the trophic information on the species and potential changes due to growth. – Reply: As indicated in the introduction (line 82: “The most drastic changes in the bay’s benthic community and food web structure 82 occurred after 2018, when the crabs reached large enough sizes to feed on the most 83 abundant organisms [15,37].”), by 2018 (when this study began) the crabs were already large enough to feed on macro and megabenthic species. However, in 2018 the benthic community did not yet drastically changed and most of the species were still present. Further in the Results the size classes are given (Table 1). Mean sizes did indeed change, but this change is between the years when the food availability has also changed. These two parameters are hard to differentiate, since in Blagopoluchiya Bay there is only one most abundant size group (Udalov et al 2024). This is a special case of crab’s population size structure, since there seems to be one settlement event that happened in 2014, and these crabs continued to grow throughout the years without any new additions (these results are discussed and presented in Udalov et al. 2024). Further, in the discussion section, this subject is further discussed: (line 422-440) “The sizes of analyzed crabs increased from 2018 to 2020 and remained within a similar range………….The sizes between 2018 and 2023 are insignificant in relation to the sizes of common macrobenthic species. It is most likely that crabs of these sizes do not have strong differences in food size preferences. Therefore, differences in their diet are more likely related to food availability.” We have attempted to clarify this argument in this discussion paragraph taking into account Your comments below.
- Results
- 220-222. In the following paragraph “…..reflects the most numerous size group sampled in the bay that year”. Which year? – Reply: we have change to “…in the bay in the respective year (Fig. 2).”, i.e. for each year given in figure 2 (sampled in the study).
- 226: "Table1" must be changed to "Table 1". – Reply: We are sorry, but we don’t understand what exactly must be changed. If this refers to the space, it is there.
- 257: The text “Further, only the results of analyzed stomachs are reported”, refers to full stomachs? – Reply: No, (line 252) “Two hundred ninety-six crabs were dissected for the stomach content analysis (Table 3), from which 260 stomachs were analyzed.” Not all dissected stomachs were analyzed, but all of them were used to calculate the proportion of fullness. This is specified in the M&M section of the manuscript (line 158) “Only empty stomachs of crabs collected in 2022 and 2023 were preserved.” We have altered this sentence to clarify: “Empty stomachs collected in 2022 and 2023 were preserved and analyzed, while those from 2018 were noted, but discarded without being analyzed.” We admit that this is a very unfortunate sampling error that adds to the unnecessary confusion in the results presentation. But we believe that (line 416) “The feeding intensity of crabs can be reflected in the amount of food in their stomachs.”, which requires all stomachs to be included in this analysis, regardless on their contents.
- 266: Use “microplasticS”. ”. – Reply: We suppose that this refers to line 265. Thank you. Corrected.
- 361-362 (and Table 6). Are means for 13C and 15N significantly different? – Reply: Purely comparing the differences in mean δ13C and δ15N separately would increase the statistical error (false positives resulting from the multiple comparisons problem) and would not account for the fact that these two values are not truly independent. Indeed, this is sometimes done in isotopic studies, when there is no possibility to perform SIBER analysis. Here we compared SEAB area (not significantly different) and their overlap (all isotopic niches overlapped in high proportion) to account for both isotopes simultaneously. And then we present the results of δ13C and δ15N comparison (398-403). This way we put the comparison of isotopic values within the framework of SIBER analysis, which hopefully decreases the likelihood of statistical error
- 402-403: “On the other hand, the δ13C was significantly different (chi-squared = 37.81, df = 7, p< 0.01) for some combinations of factors”. Please, explain. What factors would be more interesting to highlight? – Reply: The next sentence (line 403) highlights the more interesting factors “The δ 13C values of females from the Sill and males from the Inner Basin station in 2018 differed from females from both stations in 2020 (p values <0.01).”
- 406-409: This is a crucial aspect of the study. To fully illustrate this point, a figure depicting trophic ellipses for various global groups (across years/seasons) should be included, akin to Figure 7. – This is exactly the data that is reflected in the figure 7. However, your further comment and a comment from the 2nd reviewer suggest that this figure was hard to read. Therefore, we have changed it, as per suggestion of the second reviewer. Although the overlap of crabs from different stations and of different sexes is not as apparent in the new version, we believe that the overlap and differences in shape of trophic niches from different years is now clearer. Hopefully this version of the figure is more representative of the data.
Fig. 7 is difficult to analyse. I suggest increasing its width. – Reply, see reply above.
Table 1: I suggest “Range” instead of “Mean to max” (last column and table legend). – Reply: Changed
I suggest that Table 5 and Figure 6 be transferred to Supplementary Material. – Reply: Table 5 has detailed information of the results of the average food lump (AFL), and individual dominance frequency of stomach content objects, and figure 6 visually highlights the similarities between parametric SEAc and Basian SEAB calculations. These results are crucial for the argumentation in this paper, therefore, we believe they should be presented in the results section and not in the supplement material that is inconvenient for the readers. Hopefully the length of this paper allows us to keep all tables and figure. If not, we will consider restructuring these results.
- Discussion
- 419: The sentence “Crabs were collected in different months and times of day” should be considered in M&M. – Reply: we have added “different times of the day” in line 142 of the M&M section. We would also like to keep this sentence in the Discussion section, as it is crucial for further argumentation.
- 423: The authors claim that “The sizes of analyzed crabs increased from 2018 to 2020 and remained within a similar range”. What range are you referring to? – Reply: Added “of an average CW 40-50 mm (Table 1).” to the sentence.
- 430: “…most crabs were sexually mature”. Over which months did maturation take place? – Reply: unfortunately we have no idea when crabs molt in Blagopoluchiya Bay (they become sexually mature after the terminal molt). This is definitely an interesting and important question, however the remoteness of the study area does not allow us to sample crabs seasonally. Regardless of the general importance of this information for the ecological study of the crabs, we do not see how the month of molting is relevant to this study. All crabs were hard shell, so no recently molted crabs were present, that could imply fasting due to hiding. We have added the fact that all crabs had hard carapace in the results section, to, hopefully, reply to this question. Line 2016 “All crabs had hard carapace, therefore, no recently molted specimens were present.”
- 438-439: I do not understand the following statement: “The sizes between 2018 and 2023 are insignificant in relation to the sizes of common macrobenthic species”. – Reply: Indeed this is a truly unfortunate sentence. We have changed it to “Since the crabs were large enough to consume macro and megabenthic species in 2018, the increase in their sizes by 2023 is probably insignificant. It is most likely that crabs of these sizes do not have strong differences in food size preferences. Therefore, differences in their diet are more likely related to food availability.”
- 469: What kind of error are you referring to? – Reply: We mean error related to visual identification: “Visual identification of microplastics (especially transparent)”. I.E. we can’t see everything. This error is discussed in the
“Kershaw PJ, Turra A, Galgani F, GESAMP. 2019. Guidelines for the monitoring and assessment of plastic litter and microplastics in the ocean. Report. GESAMP Joint Group of Experts on the Scientific Aspects of Marine Environmental Protection. 842 https://doi.org/10.25607/OBP-435” that is cited in this statement.
- 483: “However, their quantity in full stomachs WAS greater 483 than in empty ones”. Reply: Corrected.
- 492-493. In relation to the text “This coincided with brittle stars’ dominance in these benthic communities”, understanding the sampling periods for each annual cycle is crucial. When did brittle stars’dominance occur? – Reply: We are not sure what is meant by “sampling periods for each annual cycle”. If this implies that the brittle star is only dominant in some seasons, we have no data to suggest that. They are long living organisms (more than a year) and there is no our or literature data to suggest migratory habits of brittle stars. All prior research in the Bay and in the Kara Sea in general suggest that the benthic communities were stable prior to the crab’s invasion (line 59) “The benthic communities of the Kara Sea showed undisturbed stability since the first comprehensive studies in the 20s and 30s of the XXth century that were confirmed in 2013 [21-25]” and that brittle stars were the dominant species in Blgopoluchiya Bay (lines 82-88) “The most drastic changes in the bay’s benthic community and food web structure occurred after 2018, when the crabs reached large enough sizes to feed on the most abundant organisms [15,37]. By 2020-2022, the biomass of previously abundant deposit feeders has crushed, and the most common bivalve suspension feeders diminished, leaving less attractive prey species, such as corals, sea anemones, and sponges [37]. The most dramatic change was probably in the total disappearance of previously dominant brittle stars [15].” The fact that the brittle stars were still dominant in 2018 and by 2022 they disappeared is highlighted in the next few sentences (after the sentence in question) (line 492) “Brittle stars were most common in the stomachs from the Sill station (64%) in 2018 and the Inner Basin station (60%) even in 2020. This coincided with brittle stars’ dominance in these benthic communities [15]. By 2020, the number of brittle stars had noticeably decreased [15], and they were no longer found in the crabs’ stomachs and the benthic communities in 2022 and 2023.” We have added an additional mention of the reference 15 (Udalov et al 2024) in the sentence in question. Udalov et al 2024 is a complimentary study of the Blagopoluchiya Bay benthic communities’ changes after the invasion of the crab. (co-authored by the second author of this study)
- 522: Use “in crabs’ feeding” instead of “in crabs’ nutrition”. Reply: Corrected.
- 547-549. “Crabs’ trophic position did not change over time and between the stations (TP 3.0 to 3.4), and remained at the third trophic level (secondary consumer)”. This has not been considered in M&M or Results. Also, what organism was the baseline reference in the analysis? Also, given the potential for temporal and spatial heterogeneity in baseline isotopic values, the authors should carefully consider the implications for estimating changes in a species' trophic level. This variability can introduce uncertainty into trophic position estimates. – Reply: This is indeed a statement that requires careful analysis and calculations that were performed in our different study. We have cited it in the sentence before (line 543) “The trophic position (TP) of snow crabs in Blagopoluchiya Bay was assessed in relation to other benthic species in our different work [37].» Unfortunately these two (or even three considering Udalov et al 2024 study about benthic communities changes) studies do not fit into one. However, they complement each other and it is vital to consider results from these studies for a “bigger picture” in the discussion. Therefore the trophic position of crabs is discussed here with the reference to the respective paper without giving any details about the calculations. We believe that it would be a shame to ignore trophic position in this discussion, just because it is calculated in a different study.
- 549-552. Provide references for this statement. – Reply: we are uncertain which statement requires references since both sentences in these lines have references: “Similar results were observed in the East Siberian Sea, where the δ15N values of the snow crabs are close to those of omnivorous species, which is explained by the predominant feeding on detritivores of low trophic levels (bivalves, sessile polychaetes) or feeding on detritus. In contrast, in the Chukchi Sea, the snow crabs’ δ15N values are higher, presumably due to the greater role of cannibalism [84]. In the Bering Sea, the δ15N values of the snow crabs are on par with polychaetes and brittle stars [91].”
[84] Gorbatenko KM, Kiyashko SI, Morozov TB, Glubokov AI. 2023. Distribution and General Biological Features of Snow Crab 832 (Chionoecetes opilio) in the Chukchi and East Siberian Seas. Oceanology 63(1):54-62.
[91] Kolts JM, Lovvorn JR, North CA, Grebmeier JM, Cooper LW. 2013b. Relative value of stomach contents, stable isotopes, and 846 fatty acids as diet indicators for a dominant invertebrate predator (Chionoecetes opilio) in the northern Bering Sea. Journal of 847 Experimental Marine Biology and Ecology 449:274-283.
If earlier sentences were implied (lines 544-57) then the information is based on the reference stated in the previous sentence “The trophic position (TP) of snow crabs in Blagopoluchiya Bay was assessed in relation to other benthic species in our different work [37]. “ and not given in the following two sentences to minimize repetitiveness.
[37] Zalota AK, Dgebuadz PYu, Kiselev AD, Chikina MV, Udalov AA, Kondar DV, Mishin AV, Tsurikov SM. In Press. Trophic position stability of benthic organisms in a changing food web of an Arctic fjord under the pressure of an invasive predatory snow crab, Chionoecetes opilio. Biology. Submitted.
- 596-597: A deeper focus on the dramatic shifts in macro- and megafauna populations between 2018 and 2020 is needed. What changes were observed and what were the underlying causes? – Reply: This paragraph is the conclusion of the discussion section where main ideas are summarised. We believe that more details about the changes in the benthic communities here would be excessive. Since they are given in the introduction (line 81-88) and in discussion (lines 490-509).
Regarding the underlining causes for these changes in the benthic communities of Blagopoluchiya Bay, they are not within the scope of this paper, although stated in the introduction with the reference to our other paper that is dedicated to the study of benthic communities of Blagopoluchiya Bay (line 76) “The benthic ecosystem of this bay was strongly influenced by the invasion of the snow carbs, resulting in changes in taxonomic structure, abundance, and biomass of native species [15].” This is a major study with a large amount of analysis and argumentation that could not possible fit in this paper.
- Conclusions
- 614: “The ophiuroids have disappeared from the bay” is already stated in L. 610-611. – Reply: Deleted the sentence on line 614.
- 609-617: Please, rearrange and condense the content of this section of the conclusions. – Reply: We have changed the conclusions, hopefully achieving clearer format.
Round 2
Reviewer 2 Report
Comments and Suggestions for Authors
Manuscript ID: biology-3195684
Title: Changes in the diet of an invasive predatory crab, Chionoecetes opilio, in the degrading, invaded benthic community of an Arctic fjord.
Review round 2
Answer round 1: In relation to the “most possibly, some writing program was used instead of native speaker,” comment, we would like to express our opinion.
The second author of this paper is bilingual, graduated from English school and universities, and her “native” Russian language skills are appalling. We believe that the place of birth or passport do not necessarily make “native” speakers more proficient in language skills. Many “non-native” speakers are considered great writers in languages they have studied in their life (consider Nabokov). In addition, “native” speaker professional writers (such as journalists) have several editors correcting their texts before the publication. Even great scientists are often poor writers (consider original Darwin texts). Therefore, we find it odd that in the modern “politically correct” world, where English is the most studied language, the “nativeness” of the author is used as criteria of their proficiency.
Comment: To end the discussion about "nativity", I said nothing about the things you wrote about birthplace or passport, or the proficiency of the author(s). Language proficiency is shown in your answer below. It said, "When your manuscript reaches the revision stage, you will be asked to format the manuscript according to the journal's guidelines: - You cite this, but what you did was ALREADY a revision.General comment 1 again is
Please stick to Template in revision2, and format References as required. It is NOW a revision stage.
General Comment 2:
Do not repeat data from Table in Figures. I dare to cite Editage.com: “The same data cannot be presented as both a table and a figure. This would mean duplication of content. This is one of the most basic rules of academic writing. No journal would want you to use both a table and a figure to present the same data.”
- Table 1 and Figure 2 duplicate each other and use the same data. Remove one of these.
- Table 2 is not needed here in the text, you may use asterisks or other notation to indicate significant differences in either Table 1 or Figure 2, which one you choose to not to remove.
- Figure 4 duplicates Table 5 (columns 2 to 4). It is not allowable, make revision or remove
- Figure 5 also duplicates Table 5, remove figure
If you really insist on these Figures, modify Tables accordingly. And this is not about the length of your manuscript.
General comment 3
Please refer “Figure x”, not “Fig. x” across the text. Check latest papers in the journal, if you doubt my comment.
Line 226: mistype
Reply: We are sorry, we can’t spot it. Probably we are too used to our text to spot some errors.
Now this is Line 229, “0.01, Fig. 2, Table1, 2).” Space required.
Comment round 1: Conflict of interests: check Template, and add the role of funders in …
Reply: We have added the following statement about the funding:“Both authors have received research grants from Russian Science Foundation, that does not restrict authors’ bility to analyze and interpret the data and to prepare and publish manuscripts independently when and where they choose.”
Comment: My apologies, there was no need for such an arrogant response. I did not question your skills. The question is not about the authors, but about the funders. What was really expected was the standard text based on the template, so please add this:
“The funders had no role in the design of the study, nor in the collection, analysis or interpretation of data, or the writing of the manuscript or in the decision to publish the results.”
Author Response
Dear Reviewer,
Thank you for your comments. We hope that we have fulfilled all the requirements.
Please stick to Template in revision2, and format References as required. It is NOW a revision stage.
Reply – We hope that we reformatted everything with respect to the Template.
General Comment 2:
Do not repeat data from Table in Figures. I dare to cite Editage.com: “The same data cannot be presented as both a table and a figure. This would mean duplication of content. This is one of the most basic rules of academic writing. No journal would want you to use both a table and a figure to present the same data.”
- Table 1 and Figure 2 duplicate each other and use the same data. Remove one of these.
- Table 2 is not needed here in the text, you may use asterisks or other notation to indicate significant differences in either Table 1 or Figure 2, which one you choose to not to remove.
- Figure 4 duplicates Table 5 (columns 2 to 4). It is not allowable, make revision or remove
- Figure 5 also duplicates Table 5, remove figure
If you really insist on these Figures, modify Tables accordingly. And this is not about the length of your manuscript.
Reply- We deleted Figure 2 and Table 2. Table 5 has been moved to supplement material table S2, since it does not fully duplicate figures 4, and 5, and does contain information needed for the discussion.
General comment 3
Please refer “Figure x”, not “Fig. x” across the text. Check latest papers in the journal, if you doubt my comment.
Reply – Changed.
Line 226: mistype
Reply: We are sorry, we can’t spot it. Probably we are too used to our text to spot some errors.
Now this is Line 229, “0.01, Fig. 2, Table1, 2).” Space required.
Reply – Thank you for specifying. changed.
Comment round 1: Conflict of interests: check Template, and add the role of funders in …
Reply: We have added the following statement about the funding:“Both authors have received research grants from Russian Science Foundation, that does not restrict authors’ bility to analyze and interpret the data and to prepare and publish manuscripts independently when and where they choose.”
Comment: My apologies, there was no need for such an arrogant response. I did not question your skills. The question is not about the authors, but about the funders. What was really expected was the standard text based on the template, so please add this:
“The funders had no role in the design of the study, nor in the collection, analysis or interpretation of data, or the writing of the manuscript or in the decision to publish the results.”
Reply – We were not certain how to write the information about the funders, so we copy pasted this text from the conflict of interest section in the instructions. It is unfortunate wording that sounds odd out of context. Thank you for suggesting a much better version.